# Criterion for Selective Area Growth of III-V Nanowires

**DOI:** 10.3390/nano12203698

**Published:** 2022-10-21

**Authors:** Vladimir G. Dubrovskii

**Affiliations:** Faculty of Physics, St. Petersburg State University, Universitetskaya Emb. 13B, 199034 St. Petersburg, Russia; dubrovskii@mail.ioffe.ru

**Keywords:** selective area growth, patterned substrates, openings, group III adatoms, surface diffusion

## Abstract

A model for the nucleation of vertical or planar III-V nanowires (NWs) in selective area growth (SAG) on masked substrates with regular arrays of openings is developed. The optimal SAG zone, with NW nucleation within the openings and the absence of parasitic III-V crystallites or group III droplets on the mask, is established, taking into account the minimum chemical potential of the III-V pairs required for nucleation on different surfaces, and the surface diffusion of the group III adatoms. The SAG maps are plotted in terms of the material fluxes versus the temperature. The non-trivial behavior of the SAG window, with the opening size and pitch, is analyzed, depending on the direction of the diffusion flux of the group III adatoms into or from the openings. A good correlation of the model with the data on the SAG of vertical GaN NWs and planar GaAs and InAs NWs by molecular beam epitaxy (MBE) is demonstrated.

## 1. Introduction

The III-V and III-nitride NWs, and the heterostructures within such NWs, show great promise in the fundamental studies of the physical properties of semiconductor nanomaterials and applications in nanoelectronics and optoelectronics [1,2,3,4,5,6]. Most NWs are fabricated by the vapor–liquid–solid method (with a metal catalyst droplet) [7] using different epitaxy techniques. The VLS NWs usually grow perpendicular to (111) the substrate surface [3,6,8,9,10]. Regular arrays of vertical III-V and III-nitride NWs can also be obtained by catalyst-free SAG on different masked substrates, including Si, with lithographically defined pinholes [11,12,13,14,15,16,17,18,19,20]. Furthermore, SAG enables the fabrication of template-assisted, in-plane III-V NWs, nanomembranes, nanosheets, nanofins, and advanced NW networks of different architectures [21,22,23,24,25,26,27], some of which are promising for low-temperature transport physics [23,26,27].

The SAG approach consists of the preparation of a mask on a semiconductor substrate, the lithographic patterning of the mask, and the deposition of the III-V material, which should grow only in the mask openings. The SAG approach works equally well for MBE [15,18,19,20,23,26] and metal-organic chemical vapor deposition (MOCVD) [12,13,14] techniques. Compared to other growth methods, SAG offers much wider opportunities for the geometrical design of III-V nanostructures and complex networks based on such nanostructures. In addition, the SAG safely avoids the possible contamination of the III-V materials with a growth catalyst. However, the growth selectivity is not generally guaranteed. Depending on the material combination, growth conditions and template geometry, four possible modes of SAG are possible: (i) true SAG inside the openings, without parasitic nucleation on the mask surface, (ii) random nucleation of III-V crystallites on the mask, along with single-crystalline growth inside the openings, (iii) random nucleation of group III droplets on the mask, and (iv) no growth on the mask and inside the openings [26]. Under group V-rich conditions typical for the SAG of GaN NWs, no Ga droplets can nucleate on the mask and the only possible parasitic structures are GaN crystallites or randomly oriented NWs [18,19,20]. In any case, the optimized SAG parameter window should correspond to the nucleation of the material inside the openings and the absence of parasitic nucleation on the mask [20,26]. Despite the importance of the growth selectivity issue, the comprehensive study of SAG, depending on the growth conditions and template geometry, is solely lacking in the literature. 

This work tries to fill the gap by establishing a model for the selectivity maps in terms of the material fluxes, temperature, and size and pitch of the openings. This work generalizes the earlier results obtained for the SAG of GaAs nanostructures [28,29,30], vertical GaN NWs [18,19,20], and planar GaAs and InAs NW networks [26], grown by MBE on different masked substrates. The influence of surface diffusion of group III adatoms and the minimum chemical potential required for nucleation on the growth selectivity is carefully investigated. These effects were not considered in the prior works [13,16,19,20,28,29,30]. In particular, the model of Ref. [26] assumed that desorption from the mask and semiconductor surfaces was the main mechanism determining the growth selectivity of the GaAs and InAs materials. The model of Ref. [20] was developed for the high-temperature SAG of GaN NWs without surface diffusion of the mask surface. Here, one-dimensional (1D) slits and two-dimensional (2D) circular openings (pinholes) in the mask are considered simultaneously, corresponding to the SAG of planar or vertical NWs. SAG by MBE is considered; however, the method can also be applied to MOCVD and other vapor deposition techniques. The non-trivial dependence of the SAG maps on the template geometry is studied, based on the recent results on group III adatom diffusion flux, which can be directed either from the mask surface into the openings or in the opposite direction [31,32]. The SAG windows are plotted in terms of the group III flux versus the temperature, and the group V flux versus the temperature, the V/III flux ratio versus the size and pitch of the openings and compared to the available data on vertical GaN [19,20] NWs, and planar GaAs and InAs NWs [26]. 

## 2. Model

### 2.1. SAG Criterion

A simple analysis of the initial incubation stage for the nucleation of the III-V materials inside the openings (for example, on a crystalline Si(111) surface) and on the mask (for example, on an amorphous SiO_x_ surface) is based on the following considerations. Assuming that both useful and parasitic structures are III-V crystallites, their nucleation is possible only when the chemical potential per III-V pair in the mother phase is higher than the equilibrium chemical potential [33]. For a 2D sea of group III and V adatoms adsorbed on a surface, with the surface densities $n_{3}$ and $n_{5}$, respectively, the chemical potential is given by $\mu =k_{B}Tln\left(n_{3}n_{5}\right)+f\left(T\right)$, where T is the absolute temperature, $k_{B}$ is the Boltzmann constant, and f(T) is a temperature-dependent function [33]. Therefore, the conditions for nucleation in the openings and the absence of nucleation on the mask are given by
(1)$$n_{3*}n_{5*}>n_{3*}^{eq}n_{5*}^{eq},\;n_{3}n_{5}<n_{3}^{eq}n_{5}^{eq}$$

Here, $n_{3*}n_{5*}$ refers to the openings and $n_{3}n_{5}$ to the mask surface. The $n_{3*}^{eq}n_{5*}^{eq}$ and $n_{3}^{eq}n_{5}^{eq}$ are the temperature-dependent equilibrium activities in the openings and on the mask, respectively, which should be exceeded to enable the nucleation. The SAG criterion is simply defined by the two inequalities given by Equation (1). In the incubation stage without growth, the group V atomic flux $I_{5}$ (nm^−2^s^−1^) is equalized by the desorption flux. In the case of the MBE, desorption occurs in the form of dimers (As_2_, P_2_ or N_2_) [20]. Therefore, one can write $I_{5}=2D_{5*}n_{5*}^{2}=2D_{5}n_{5}^{2}$, with $D_{5*}$ and $D_{5}$ as the effective diffusion coefficients of the group V adatoms in the openings and on the mask, yielding
(2)$$n_{5*}=\sqrt{I_{5}/\left(2D_{5*}\right)},\;n_{5}=\sqrt{I_{5}/\left(2D_{5}\right)}$$

In MBE, the group V adatom surface densities scale as $I_{5}^{1/2}$ [20]. In MOCVD, the group V element may desorb either in the form of dimers or as a precursor molecule, such as AsH_3_. In the latter case, both of the group V adatom surface densities are proportional to $I_{5}$. Without surface diffusion of the group III adatoms into or from the openings, the group III adatom surface densities are given by [33]
(3)$$n_{3*}=I_{3}\tau_{3*},n_{3}=I_{3}\tau_{3}$$

Here, $I_{3}$ is the atomic flux of the group III element; $\tau_{3*}$ is the mean lifetime of the group III adatoms in the openings before desorption (the desorption time for brevity); and $\tau_{3}$ is the desorption time on the mask.

Using Equations (2) and (3), Equation (1) can be put as
(4)$$\frac{n_{3*}^{eq}n_{5*}^{eq}}{\tau_{3*}}\sqrt{\frac{2D_{5*}}{I_{5}}}<I_{3}<\frac{n_{3}^{eq}n_{5}^{eq}}{\tau_{3}}\sqrt{\frac{2D_{5}}{I_{5}}}$$

These inequalities show that the group III flux in the optimized SAG regime should be neither too low to ensure nucleation in the openings nor too high to suppress parasitic nucleation on the mask, as discussed in Refs. [16,19,20,26,28,29,30] for the SAG of GaAs, InAs, and GaN materials on different masked substrates. The standard Arrhenius temperature dependences of the parameters entering Equation (4) are given by [33]: $n_{3}^{eq}n_{5}^{eq}\propto \exp\left(-\Lambda_{35}/k_{B}T\right)$, $1/\tau_{3}\propto exp\left(-E_{3}^{des}/k_{B}T\right)$, and $D_{5}\propto exp\left(-E_{5}^{diff}/k_{B}T\right)$, with $\Lambda_{35}$ as the specific condensation heat of the III-V material; $E_{3}^{des}$ as the activation energy for the desorption of the group III adatom from the mask surface; and $E_{5}^{diff}$ as the activation energy of the dimerization of the group V adatoms on the mask, and similar expressions for the openings. Using these temperature dependences in Equation (4), one arrives at
(5)$$\frac{C_{*}e^{-E_{*}/k_{B}T}}{I_{5}^{1/2}}<I_{3}<\frac{Ce^{-E/k_{B}T}}{I_{5}^{1/2}}$$
with the effective activation energies $E_{*}=\Lambda_{35*}+E_{3*}^{des}+E_{5*}^{diff}/2$ in the openings and $E=\Lambda_{35}+E_{3}^{des}+E_{5}^{diff}/2$ on the mask. The constants $C_{*}$ and C (nm^−1^s^−1/2^) summarize the pre-exponential factors in the Arrhenius temperature dependences of the different parameters. According to this criterion, SAG without a parasitic nucleation of III-V crystallites on the mask requires that $I_{3}$ is within a certain range. The minimum value of the group III flux separates the SAG zone from the no-growth zone, while the maximum group III flux separates the SAG zone from the zone of the parasitic nucleation on the mask. Both of the critical fluxes increase with temperature, following the Arrhenius dependence and decrease, with the group V flux as $I_{5}^{-1/2}$.

For high-enough group III fluxes, the parasitic III-V crystallites on the mask can be replaced by liquid or partly solidified Ga-rich droplets, whose group V content is close to equilibrium at a given temperature [26]. At $n_{5}=n_{5}^{eq}$, the upper limit for the chemical potential on the mask changes to
(6)$$n_{3}<n_{3}^{eq}$$
with $n_{3}^{eq}$ as the equilibrium concentration of the group III adatoms, which should be exceeded to nucleate a droplet. In this case, the SAE criterion changes to
(7)$$\frac{C_{*}e^{-E_{*}/k_{B}T}}{I_{5}^{1/2}}<I_{3}<Ae^{-E_{1}/k_{B}T}$$
where the upper limit for the group III flux is independent of $I_{5}$. The effective activation energy, $E_{1}$, is given by $E_{1}=\Lambda_{3}+E_{3}^{des}$, where $\Lambda_{3}$ is the specific condensation heat for the group III droplets emerging on the mask. 

The criteria given by Equation (5) or (7) are not very different from those obtained in Ref. [20] for the SAG-MBE of vertical GaN NWs on patterned SiO_x_/Si(111) substrates with regular arrays of pinholes, and Ref. [26] for the planar GaAs and InAs NW networks on patterned SiO_x_/GaAs(100) substrates. In Ref. [26], the authors considered desorption from the openings or the mask as the major process governing the presence or absence of nucleation. The considerations presented here are slightly more complex, as they account for the minimum chemical potential required for nucleation and include the group V flux. 

### 2.2. Surface Diffusion of Group III Adatoms

In a more complex situation with surface diffusion, one needs to consider the diffusion-induced exchange of the group III adatoms between the openings and the mask [16,32,33]. The geometry of the regular arrays of 1D or 2D openings is illustrated in Figure 1, along with the model parameters. As above, the III-V material is deposited from the atomic fluxes, $I_{3}$ and $I_{5}$. The effective lifetimes of the group III adatoms in the openings ($\tau_{*}$) and on the mask (τ) account for the nucleation of III-V crystallites, as specified below. The rate constants for the transfers of the group III adatoms from the mask to the openings ($k^{+},$ nm/s), and from the openings to the mask ($k^{-}$, nm/s), are assumed independent of the surface densities, as in Ref. [31] and [32]. In what follows, I will use 2D geometry of the regular circular openings (pinholes) in the mask layer, relevant for the SAG of the vertical NWs [2,11,12,13,14,15,16,17,18,19,20]. The corresponding results for the 1D slits, relevant for the SAG of the in-plane NWs [24,25,26] or nanomembranes [21,22], are easily obtained from the 2D case, as explained below.

In the 2D case, the pitch P and pinhole radius R are related as P=L+2R, where L is the distance from the central band, between the pinholes, to the hole periphery. The coordinate-dependent adatom density on the mask obeys the stationary 2D diffusion equation
(8)$$D_{3}\Delta n_{3}+I_{3}-\frac{n_{3}}{\tau_{3}}-D_{3}\left(n_{3}n_{5}-n_{3}^{eq}n_{5}^{eq}\right)=0$$
with $D_{3}$ as the diffusion coefficient on the mask surface. Here, the sink, $D_{3}\left(n_{3}n_{5}-n_{3}^{eq}n_{5}^{eq}\right)$, accounts for the possible nucleation of the III-V material on the mask [34,35] in the simplest approximation, which is linear in $n_{3}$, similarly to Ref. [32]. This term is effective only when $n_{3}n_{5}>n_{3}^{eq}n_{5}^{eq}$. Equation (8) can be re-written in the form
(9)$$D_{3}\Delta n_{3}+A-\frac{n_{3}}{\tau }=0$$
with
(10)$$A=I_{3}+D_{3}n_{3}^{eq}n_{5}^{eq},\frac{1}{\tau }=\frac{1}{\tau_{3}}+D_{3}n_{5}$$

The boundary conditions to Equation (9) are given by
(11)$${\left(\frac{dn_{3}}{dr}\right)}_{r=0}=0,-D_{3}{\left(\frac{dn_{3}}{dr}\right)}_{r=L/2}=k^{+}n_{3}\left(r=L/2\right)-k^{-}n_{3*}$$

The first boundary condition corresponds to a zero-diffusion flux between the pinholes, neglecting the substrate edges. The second boundary condition, at the mask-opening interface, is obtained from the transition rate theory [31,32,36], used previously in the field of NW growth [37], with the corresponding rate constants, $k^{+}$ and $k^{-}$. To find the unknown surface density in the slits $n_{3*}$, a third boundary condition is required. From the material balance considerations, it follows that
(12)$$2\pi RD_{3}{\left(\frac{dn_{3}}{dr}\right)}_{r=L/2}=\pi R^{2}\left[I_{3}-\frac{n_{3*}}{\tau_{3*}}-D_{3*}\left(n_{3*}n_{5*}-n_{3*}^{eq}n_{5*}^{eq}\right)\right]$$

According to this equation, the difference between the deposition flux of the group III atoms and their sink, due to desorption and nucleation, equals the total diffusion current through the mask-opening interface. This current can be positive, corresponding to a diffusion of group III adatoms from mask to openings, or negative otherwise [31,32]. Although it is not critical, we assume a spatially uniform adatom density in the openings ($n_{3*}=const$), which is relevant when the size of the openings is much smaller than the pitch (or when the surface diffusivity on the mask is much lower than on the substrate surface in the openings). The non-uniform adatom density in the openings in the 1D case has recently been considered in Ref. [32].

The solutions for the adatom surface densities are given by
(13)$$n_{3*}=A_{*}\tau_{*}\frac{\left(A/A_{*}\tau_{*}\right)k^{+}\tau /\lambda_{3}+y}{k^{-}\tau_{*}/\lambda_{3}+y},\;y=\frac{R}{2\lambda_{3}}\left[1+\frac{k^{+}\tau }{\lambda_{3}}\frac{i_{0}\left(L/2\lambda_{3}\right)}{i_{1}\left(L/2\lambda_{3}\right)}\right]$$
with
(14)$$A_{*}=I_{3}+D_{3*}n_{3*}^{eq}n_{5*}^{eq},\frac{1}{\tau_{*}}=\frac{1}{\tau_{3*}}+D_{3*}n_{5*}$$
in the openings, and
(15)$$n_{3}=A\tau +\frac{k^{-}n_{3*}-k^{+}A\tau }{k^{+}i_{0}\left(L/2\lambda_{3}\right)+\left(\lambda_{3}/\tau \right)i_{1}\left(L/2\lambda_{3}\right)}i_{0}\left(r/\lambda_{3}\right)$$
on the mask. Here, $i_{0}$ and $i_{1}$ denote the Bessel functions of the first kind of order 0 and 1, respectively. The effective diffusion length of the group III adatoms on the mask surface $\lambda_{3}=\sqrt{D_{3}\tau }$ can be limited either by desorption or nucleation and the growth of parasitic III-V crystallites. These solutions enable the analysis of the diffusion fluxes and adatom surface densities with nucleation and growth, as in Ref. [32]. In the linear approximation considered here, the nucleation dynamics are driven by the chemical potential difference described by the terms $n_{3}n_{5}-n_{3}^{eq}n_{5}^{eq}$ and $n_{3*}n_{5*}-n_{3*}^{eq}n_{5*}^{eq}$ in the diffusion equations. Nucleation is expected to be faster at lower temperatures (due to reduced $n_{3}^{eq}n_{5}^{eq}$) and higher material fluxes (due to larger $n_{3}n_{5}$). The growth modeling of large crystallites inside the openings requires an additional analysis of the three-dimensional (3D) growth, and the surface energies of different structures and their geometry, which is beyond the scope of this work but will be considered elsewhere.

The following analysis is restricted to the incubation stage, where $A=A_{*}=I_{3}$, $\tau =\tau_{3}$, $\tau_{*}=\tau_{3*}$, and $\lambda_{3}=\sqrt{D_{3}\tau_{3}}$ is the desorption-limited diffusion length. This analysis corresponds to the incomplete condensation regime with slow nucleation, where the supersaturation increases to a maximum, determined by the adsorption–desorption balance, and then slowly decreases due to the nucleation and growth of the III-V crystallites [33]. In this case, the above solutions for the adatom surface densities are reduced to
(16)$$n_{3*}=I_{3}\tau_{3*}\frac{k^{+}\tau_{3}/\lambda_{3}+y}{k^{-}\tau_{3*}/\lambda_{3}+y},\;y=\frac{R}{2\lambda_{3}}\left[1+\frac{k^{+}\tau_{3}}{\lambda_{3}}\frac{i_{0}\left(L/2\lambda_{3}\right)}{i_{1}\left(L/2\lambda_{3}\right)}\right]$$
(17)$$n_{3}=I_{3}\tau_{3}\left[1+\frac{\left(k^{-}\tau_{3*}-k^{+}\tau_{3}\right)}{\lambda_{3}}\frac{i_{0}\left(r/\lambda_{3}\right)}{(1+2k^{-}\tau_{3*}/R)i_{1}\left(L/2\lambda_{3}\right)+\left(k^{+}\tau_{3}/\lambda_{3}\right)i_{0}\left(L/2\lambda_{3}\right)}\right]$$

At $k^{+}\tau_{3}>k^{-}\tau_{3*}$, the diffusion flux of the group III adatoms is directed from the mask surface into the pinholes, as in Refs. [31,32]. From Equation (16), the adatom density in the pinholes $n_{3*}$ is larger than $I_{3}\tau_{3*}$. From Equation (17), the adatom density on the mask is smaller than $I_{3}\tau_{3}$ and decreases from the center toward the pinhole periphery. At $k^{+}\tau_{3}<k^{-}\tau_{3*}$, the diffusion flux is directed from the pinholes onto the mask surface [31,32]. In this case, $n_{3*}<I_{3}\tau_{3*}$, while $n_{3}>I_{3}\tau_{3}$ and increases from the center toward the pinhole periphery. These different behaviors of $n_{3}$ are illustrated in Figure 2. 

For the coordinate-dependent adatom density on the mask, the second inequality in the SAG criterion should be modified to
(18)$$n_{3,max}n_{5}<n_{3}^{eq}n_{5}^{eq}$$
meaning that parasitic nucleation is not possible anywhere on the mask surface. Using the obtained solutions, the SAG criterion takes the form
(19)$$\frac{C_{*}e^{-E_{*}/k_{B}T}}{I_{5}^{1/2}}\frac{k^{-}\tau_{3*}/\lambda_{3}+y}{k^{+}\tau_{3}/\lambda_{3}+y}<I_{3}<\frac{Ce^{-E/k_{B}T}}{I_{5}^{1/2}}\frac{1}{1+\left(k^{-}\tau_{3*}-k^{+}\tau_{3}\right)z/\lambda_{3}}$$

Here, the function z is different for the different directions of the diffusion flux:(20)$$\begin{array}{c} z=\frac{1}{(1+2k^{-}\tau_{3*}/R)i_{1}\left(L/2\lambda_{3}\right)+\left(k^{+}\tau_{3}/\lambda_{3}\right)i_{0}\left(L/2\lambda_{3}\right)},\;k^{+}\tau_{3}>k^{-}\tau_{3*}\\ z=\frac{1}{(1+2k^{-}\tau_{3*}/R)i_{1}\left(L/2\lambda_{3}\right)/i_{0}\left(L/2\lambda_{3}\right)+k^{+}\tau_{3}/\lambda_{3}},\;k^{+}\tau_{3}<k^{-}\tau_{3*} \end{array}$$

In the 1D geometry of the regular slits, with the width W and pitch P, the opening radius R should be changed to W; $i_{0}\left(L/2\lambda_{3}\right)$ should be changed to $cosh\left(L/2\lambda_{3}\right)$; and $i_{1}\left(L/2\lambda_{3}\right)$ changed to $sinh\left(L/2\lambda_{3}\right)$, with L=P−W. Therefore, the SAG criterion for the 1D slits is given by the same Equation (19), with
(21)$$y=\frac{W}{2\lambda_{3}}\left[1+\frac{k^{+}\tau_{3}}{\lambda_{3}}cotanh\left(\frac{L}{2\lambda_{3}}\right)\right]$$
and
(22)$$\begin{array}{c} z=\frac{1}{(1+2k^{-}\tau_{3*}/R)sinh\left(L/2\lambda_{3}\right)+\left(k^{+}\tau_{3}/\lambda_{3}\right)cosh\left(L/2\lambda_{3}\right)},\;k^{+}\tau_{3}>k^{-}\tau_{3*}\\ z=\frac{1}{(1+2k^{-}\tau_{3*}/R)cotanh\left(L/2\lambda_{3}\right)+k^{+}\tau_{3}/\lambda_{3}},\;k^{+}\tau_{3}<k^{-}\tau_{3*} \end{array}$$

The SAG criterion, given by Equation (19), with the geometry-dependent functions y and z, given by Equations (16) and (20) in the 2D geometry, or Equations (21) and (22) in 1the D geometry, is the main result of this work. In the next section, the SAG window is analyzed as a function of the temperature, material fluxes, opening size, and pitch, and compares the results to some experimental data. 

## 3. Results and Discussion

The temperature dependence of the SAG window is given by the Arrhenius exponents in Equation (19), the temperature-dependent values of $k^{+}\tau_{3}$ and $k^{-}\tau_{3*}$, and the temperature-dependent diffusion length $,\;\lambda_{3}$, in Equation (16) or (21) for y and Equation (20) or (22) for z. The temperature behavior of the rate constants, $k^{+}$ and $k^{-}$, is generally unknown. The diffusion length, $\lambda_{3}$, must decrease with the temperature due to the enhanced desorption. Both desorption times, $\tau_{3}$ and $\tau_{3*}$, decrease with higher temperatures. One should anticipate $k^{+}$ to be larger than $k^{-}$, due to the presence of the Schwoebel barrier for the diffusion escape of the group III adatoms from the pinholes [38]. This barrier should increase with the depth of the holes. On the other hand, the group III atoms should be weaker bound to the inert mask surface than to the crystalline substrate in the pinholes, which is why the desorption time on the mask, $\tau_{3}$, should be shorter or even much shorter than $\tau_{3*}$. This may result in the negative difference $k^{+}\tau_{3}-k^{-}\tau_{3*}$, even at $k^{+}>k^{-}$, as suggested in Ref. [32], to explain the counter-intuitive behavior of the GaAs’ growth rate with the pinhole size and pitch. The desorption times, $\tau_{3}$ and $\tau_{3*}$, depend differently on temperature. This may change the direction of the group III diffusion flux, probably, from positive to negative with the increasing temperature. Very high growth temperatures should result, however, in the suppression of surface diffusion. At $\;k^{-}\tau_{3*}/\lambda_{3}\ll 1$ and $k^{+}\tau_{3}/\lambda_{3}\ll 1$, the SAG criterion given by Equation (19) is reduced to Equation (5), obtained by neglecting the surface diffusion. 

For example, the SAG of the GaN NWs on the patterned SiO_x_/Si(111) substrates was performed by plasma-assisted MBE (PA-MBE) within a high temperature range, from 810 °C to 850 °C [20]. The pinhole pitch was fixed at 1600 nm, and the pinhole radius was 225 nm. The growth selectivity window was studied experimentally for different Ga/N_2_ flux ratios. The flux ratio was varied by changing the Ga flux at a fixed N_2_ flux, corresponding to an N_2_ flow of 0.4 sccm. Without surface diffusion, Equation (5) gives the SAG window in terms of the V/III flux ratio in the form
(23)$$\frac{C_{*}e^{-E_{*}/k_{B}T}}{I_{5}^{3/2}}<\frac{I_{3}}{I_{5}}<\frac{Ce^{-E/k_{B}T}}{I_{5}^{3/2}}$$

Figure 3 shows the SAG window for the GaN NWs at a fixed N_2_ flow of 0.4 sccm, obtained from this expression with $C_{*}/I_{5}^{3/2}=$ 1.25 × 10^24^, $C/I_{5}^{3/2}=$ 3.65 × 10^2^, $E_{*}=$ 5.69 eV, and E= 5.08 eV. The theoretical curves fit the data very well and separate the optimal SAG window from the zone of the parasitic growth of GaN crystallites on the mask for the higher Ga fluxes and the no-growth zone for the lower Ga fluxes. Decreasing the N_2_ flow from 0.4 sccm to 0.2 sccm results in the higher Ga fluxes or Ga/N_2_ ratios required for the SAG, due to the presence of the $I_{5}^{3/2}$ denominators in Equation (23).

The networks of the planar GaAs and InAs NW of Ref. [26] were grown by MBE on different substrates, including GaAs(100) and GaAs(111)B, in the openings of the SiO_x_ masks with different configurations. Figure 4 shows the SAG windows for the homoepitaxial growth of the GaAs on GaAs(100), in terms of the As and Ga fluxes versus the temperature. Figure 5 shows the SAG windows for the heteroepitaxial growth of the InAs on GaAs(100), in terms of the As and In fluxes versus the temperature.

As discussed above, the III/V flux ratios in these SAGs are more balanced than in the case of the GaN NWs shown in Figure 3. Therefore, the enhanced group III fluxes result in the parasitic nucleation of Ga or In droplets, saturated with As [26]. In this case, the maximum group III flux is given by Equation (7) and is independent of $I_{5}$. The two inequalities for the minimum and maximum group V fluxes, at a given $I_{3}$, are easily obtained from Equation (5). The surface diffusion of the Ga or In adatoms into or from the openings should be effective within the temperature domains employed for the SAG. However, nothing can be said about this contribution without measuring the nucleation or growth rates of the III-V material as a function of the slit size and pitch, as in Ref. [32]. Therefore, the data are fitted using the Arrhenius temperature dependences given by
(24)$$A_{*}e^{-2E_{*}/k_{B}T}<I_{5}<Ae^{-2E/k_{B}T},B_{*}e^{-E_{1*}/k_{B}T}<I_{3}<Be^{-2E_{1}/k_{B}T}$$
where the pre-exponential factors may contain the geometry-dependent diffusion terms. The form of these expressions is not different from the simplified desorption rates considered in Ref. [26]. 

Figure 4 shows the measured critical As and Ga fluxes for the homoepitaxial SAG of the GaAs on the GaAs(100) substrates separating the SAG zone from the no-growth and parasitic growth domains [26], fitted by Equation (24). The minimum As flux in Figure 4a was obtained with $A_{*}$ corresponding to 7.0 × 10^12^ ML/s and $2E_{*}=$ 2.16 eV. The maximum As flux in Figure 4a was obtained with A corresponding to 8.0 × 10^15^ ML/s and 2E= 3.06 eV. The minimum Ga flux in Figure 4b was obtained with $B_{*}$ corresponding to 1.15 × 10^13^ ML/s and $E_{1*}=$ 2.40 eV. The maximum Ga flux in Figure 4b was obtained with B corresponding to 1.12 × 10^14^ ML/s and $E_{1}=$ 2.85 eV. Figure 5 shows the measured critical As and In fluxes for the heteroepitaxial SAG of the InAs on the GaAs(100) substrates separating the SAG [26] and their fits by Equation (24). The minimum As flux in Figure 5a was obtained with $A_{*}$ corresponding to 9.5 × 10^14^ ML/s and $2E_{*}=$ 2.16 eV. The maximum As flux in Figure 5a was obtained with A corresponding to 3.65 × 10^13^ ML/s and 2E= 2.33 eV. The minimum In flux in Figure 5b was obtained with $B_{*}$ corresponding to 8.5 × 10^7^ ML/s and $E_{1*}=$ 1.28 eV. The maximum In flux in Figure 5b was obtained with B corresponding to 2.8 × 10^11^ ML/s and $E_{1}=$ 2.15 eV. The activation energies for both critical Ga fluxes appear higher than those for the In flux, which seems plausible. The good fits to the data confirm that the Arrhenius-like temperature behavior of the SAG windows is predominant for these material systems under the growth conditions and for the pattern geometries employed in Ref. [26]. The activation energies obtained from the fits in Figure 4 and Figure 5 are very close to those given in Ref. [26], with only one exception, in Figure 4a, where our fit gives a lower activation energy for the minimum As flux, of 2.16 eV compared to 2.5 eV in Ref. [26].

Very importantly, the SAG windows depend on the template geometry. An analysis of the SAG criterion with surface diffusion, given by Equation (19), with the functions y and z, given by Equations (16) and (20), respectively, leads to the following conclusions. When the diffusion flux of the group III adatoms is directed from the mask to the openings (at $k^{+}\tau_{3}>k^{-}\tau_{3*}$), the minimum group III flux or III/V flux ratio separating the SAG zone from the no-growth zone is smaller than the minimum flux without surface diffusion. The minimum group III flux increases with the opening size and decreases with the pitch. The maximum group III flux or III/V flux ratio separating the SAG zone from the parasitic growth on the mask is larger than the maximum flux without surface diffusion. The maximum flux increases with the opening size and decreases with the pitch. When the group III adatoms diffuse from the openings onto the mask surface (at $k^{+}\tau_{3}<k^{-}\tau_{3*})$, the situation is fully reversed. The SAG zone narrows with respect to the idealized case without surface diffusion. Both the minimum and maximum group III fluxes or III/V flux ratios decrease with the opening size and increase with the pitch. 

Figure 6 shows the examples of SAG windows in terms of the III/V flux ratio versus the dimensionless radius, $R/\lambda_{3}$, at a fixed pitch of the 2D pinholes (the obtained solutions to the linear diffusion equations for the adatom density contain only the dimensionless radius, $R/\lambda_{3}$, and the pitch, $P/\lambda_{3}$). The curves are obtained from Equations (16), (19), and (20), with the same parameters as for the vertical GaN NWs on the SiO_x_/Si(111) substrates at a fixed temperature of 830 °C, $k^{+}\tau_{3}/\lambda_{3}=1$, and different $k^{-}\tau_{3*}/\lambda_{3}$, corresponding to the surface diffusion from the mask into the pinholes (modelled with $(k^{+}\tau_{3}-k^{-}\tau_{3*})/\lambda_{3}=0.9$), or, in the opposite direction (modelled with $(k^{+}\tau_{3}-k^{-}\tau_{3*})/\lambda_{3}=-0.5$), compared to the case without surface diffusion (at $k^{+}\tau_{3}=k^{-}\tau_{3*}$). It is seen that the SAG window is much wider for positive diffusion from the mask into the openings. This effect is due to an increase in the adatom density in the pinholes and its decrease on the mask. Lower group III fluxes or III/V flux ratios are required for SAG in smaller pinholes. For negative surface diffusion from the pinholes onto the mask surface, the SAG window is much narrower because such diffusion decreases the adatom density in the pinholes and increases it on the mask. Higher group III fluxes or III/V flux ratios are required for SAG in smaller pinholes.

Figure 7 shows the SAG windows for the vertical NWs in terms of the III/V flux ratio versus the dimensionless pitch, $P/\lambda_{3}$, at a fixed radius of 2D pinholes, for the same parameters as in Figure 6. The SAG window is again wider for the positive diffusion flux into the pinholes. Higher group III fluxes are required to grow NWs in denser pinholes. This effect is associated with a competition of the different openings for the group III diffusion flux from the mask surface [16]. For the SAG of the vertical GaN NWs on the patterned SiO_2_/Si(111) substrates by PA-MBE [19], too sparse pinhole patterns resulted in an extensive parasitic nucleation around the pinholes, while too dense patterns frequently resulted in pinholes which remained unoccupied. These features can qualitatively be understood from the curves shown in Figure 7. At a fixed group III flux, larger pitches may result in parasitic nucleation on the mask at $I_{3}>I_{3max}$, while smaller pitches may suppress GaN nucleation in the pinholes at $I_{3}<I_{3min}$, with $I_{3max}$ and $I_{3min}$ corresponding to the maximum and minimum group III fluxes separating the SAG domain. The SAG window narrows when the group III diffusion flux is directed from the pinholes onto the mask. In contrast to the previous case, lower group III fluxes are required for the SAG in denser pinholes. According to Figure 6 and Figure 7, the SAG of the NWs becomes more difficult when the group III adatoms diffuse from the openings onto the mask surface. When this diffusion flux gets very high, the SAG windows may completely disappear.

## 4. Conclusions

In conclusion, our theoretical analysis leads to the following new results. The group III fluxes or III/V flux ratios relevant for SAG are restricted by the two critical curves. The minimum curve separates the SAG zone from the no-growth zone, while the maximum curve corresponds to the onset of the parasitic nucleation of III-V crystallites or group III droplets on the mask. The minimum group III flux corresponds to the temperature-dependent equilibrium chemical potential per III-V pair in the openings, which should be exceeded to nucleate the III-V material on a semiconductor surface. The maximum group III flux corresponds to the equilibrium chemical potential per III-V pair on the mask, which should not be exceeded to suppress parasitic nucleation. Both chemical potentials increase with temperature, which explains why the group III and group V fluxes required for SAG are larger for higher temperatures. The chemical potentials increase with the group V flux due to a higher surface density of the group V atoms. Consequently, the higher group V fluxes reduce the optimal group III fluxes in the SAG domain. The Arrhenius temperature dependence of the SAG windows fits quite well the data on the SAG of the vertical GaN NWs, planar GaAs, and InAs NWs by MBE on the different substrates covered with the SiO_x_ masks. It is important, however, that the activation energies in these Arrhenius curves contain the characteristics of desorption, surface diffusion, and dimerization of the group III and group V atoms, rather than just desorption [26]. Indeed, any material influxes lead to non-vanishing concentrations of adatoms, which may be higher or lower than the equilibrium values, to enable or suppress the surface nucleation [33,35]. 

The influence of template geometry on the SAG windows originates from the surface diffusion of the group III adatoms, and critically depends on the direction of the diffusion flux into or from the openings. Surface diffusion from the mask into the openings yields wider SAG windows for the material fluxes and temperature. Higher group III fluxes are required to grow NWs in larger openings with smaller pitches. This behavior is reversed for the opposite direction of the diffusion flux, where the SAG windows are significantly narrowed. A physical interpretation of this narrowing effect is quite clear. A positive diffusion flux of group III adatoms from the mask to the openings increases the chemical potential in the openings and decreases it on the mask surface, yielding a wider range of SAG conditions. A negative diffusion flux decreases the chemical potential in the openings and increases it on the mask surface, which narrows the SAG window. For the same reason, smaller openings and larger pitches extend the SAG window for a positive diffusion flux and narrow it for a negative diffusion flux. These findings should be useful for selecting the correct conditions for SAG in a wide range of material systems using templates of different architecture. Similar approaches may be used for the modeling of the SAG in the later stages, including the nucleation and growth dynamics of 3D crystallites.

## Figures and Tables

**Figure 1 nanomaterials-12-03698-f001:**
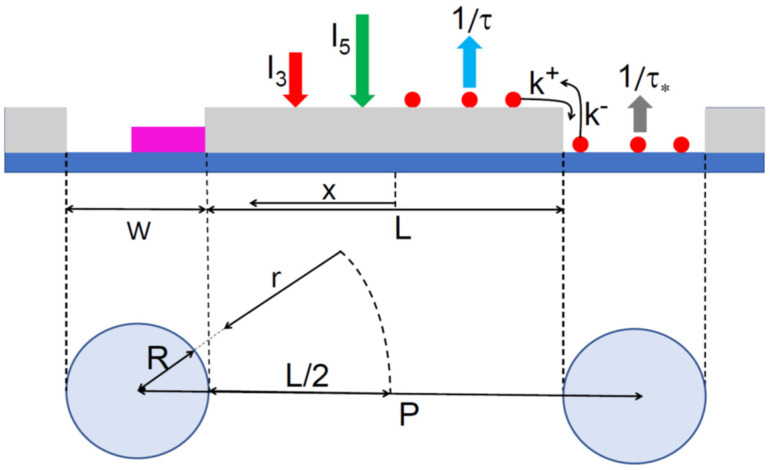
Geometry of regular 1D slits of width W, or 2D arrays of pinholes of radius R, separated by pitch P. The coordinates x in 1D geometry and r  in 2D geometry originate from the central band between the openings. The III-V material is deposited from atomic fluxes of group III and V elements $I_{3}$ and $I_{5}$. The effective lifetimes of group III adatoms in the openings and on the mask are $\tau_{*}$ and τ, respectively. The rate constants for the transfers of group III adatoms from mask to openings and from openings to mask are $k^{+}$ and $k^{-}$, respectively.

**Figure 2 nanomaterials-12-03698-f002:**
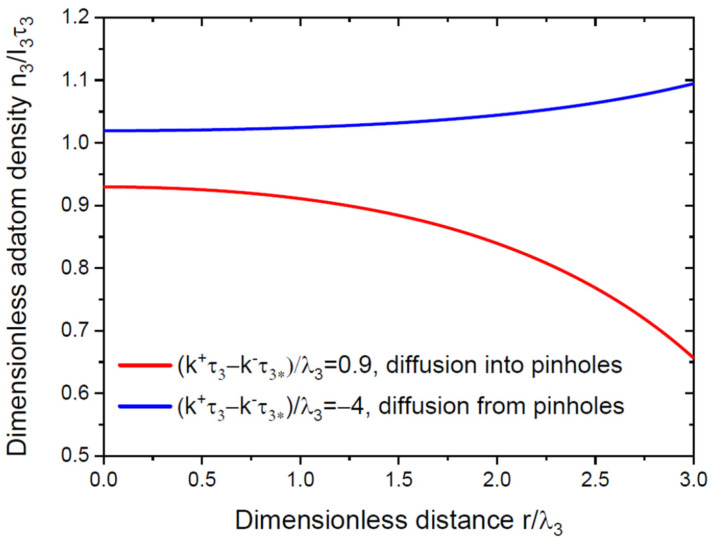
Dimensionless group III adatom density on the mask versus dimensionless distance $r/\lambda_{3}$, decreasing toward the pinhole periphery at $k^{+}\tau_{3}>k^{-}\tau_{3*}$, and increasing otherwise.

**Figure 3 nanomaterials-12-03698-f003:**
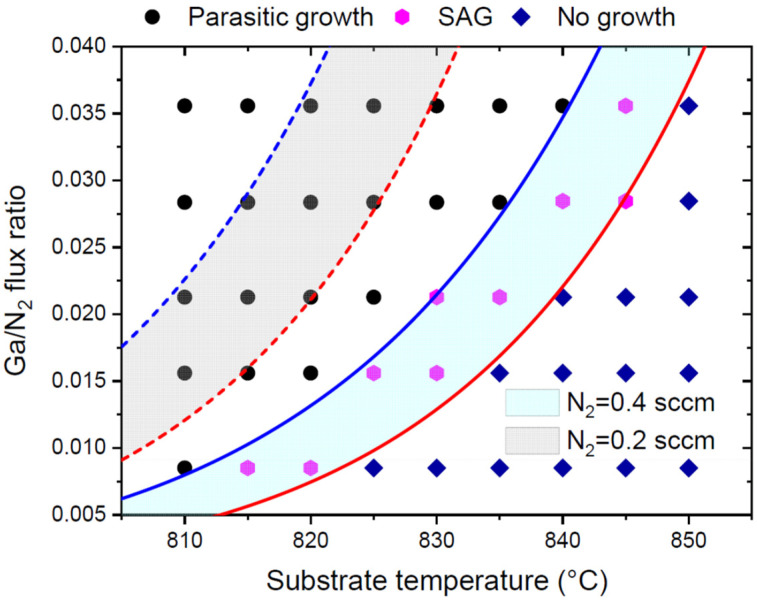
SAG windows for vertical GaN NWs grown by PA-MBE on patterned SiO_x_/Si(111) in terms of temperature-dependent Ga/N_2_ flux ratios at two different N_2_ flows shown in the legend. The curves are obtained from Equation (23) without surface diffusion. Symbols correspond to parasitic growth, true SAG, and no-growth conditions observed experimentally for N_2_ flux corresponding to 0.4 sccm [20].

**Figure 4 nanomaterials-12-03698-f004:**
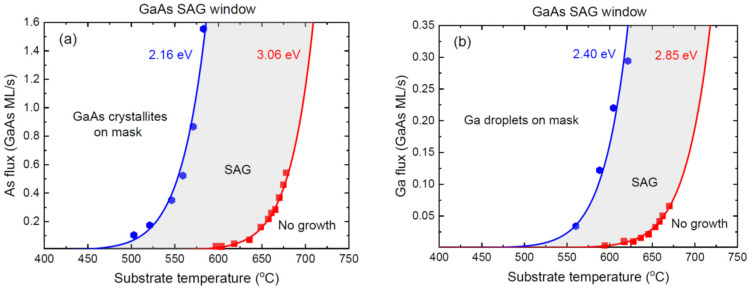
SAG windows for homoepitaxial GaAs SAG on GaAs(100): (**a**) As flux and (**b**) Ga flux versus temperature. Symbols represent the datapoints of Ref. [26], separating the SAG zone from the domains of parasitic nucleation of GaAs crystallites in (**a**) or Ga droplets in (**b**) when As or Ga fluxes are too high, and the no growth conditions for low fluxes. The lines correspond to the maximum and minimum fluxes obtained from Equation (24) with the effective activation energies shown near each curve.

**Figure 5 nanomaterials-12-03698-f005:**
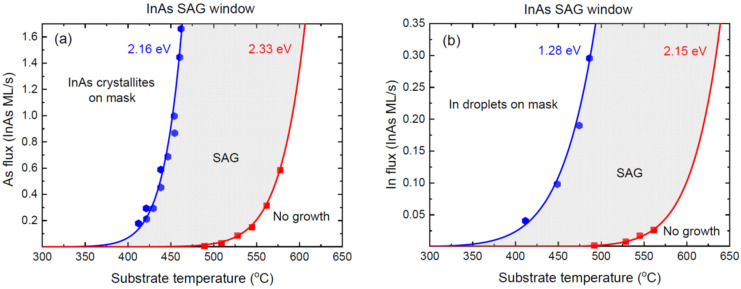
Same as Figure 4 for heteroepitaxial InAs SAG on GaAs(100). Symbols represent the datapoints of Ref. [26]. The lines correspond to the maximum and minimum fluxes obtained from Equation (24) with the effective activation energies shown near each curve.

**Figure 6 nanomaterials-12-03698-f006:**
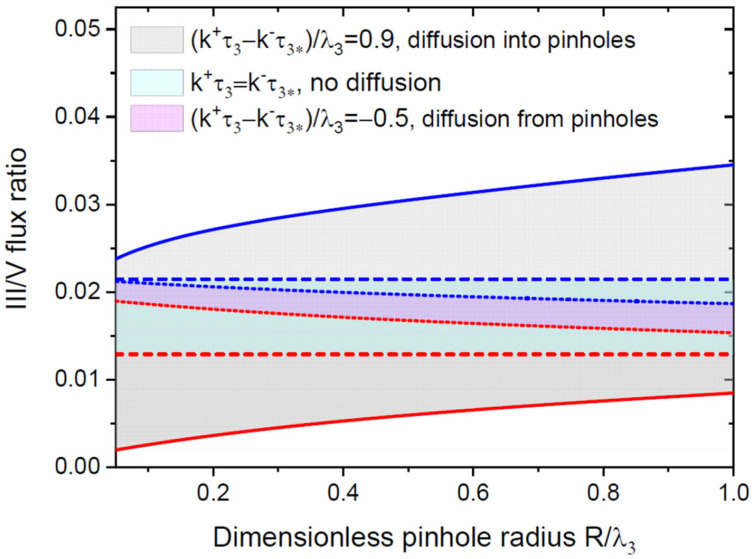
SAG windows for vertical NWs in terms of III/V flux ratio versus dimensionless pinhole radius $R/\lambda_{3\;}$ in the case of positive (solid lines), zero (dashed lines), and negative (dotted lines) diffusion flux of group III adatoms from the mask surface into the pinholes, corresponding to different values of $k^{+}\tau_{3}-k^{-}\tau_{3*}$ shown in the legend. The dimensionless pinhole pitch $P/\lambda_{3\;}\;$ is fixed at 4.

**Figure 7 nanomaterials-12-03698-f007:**
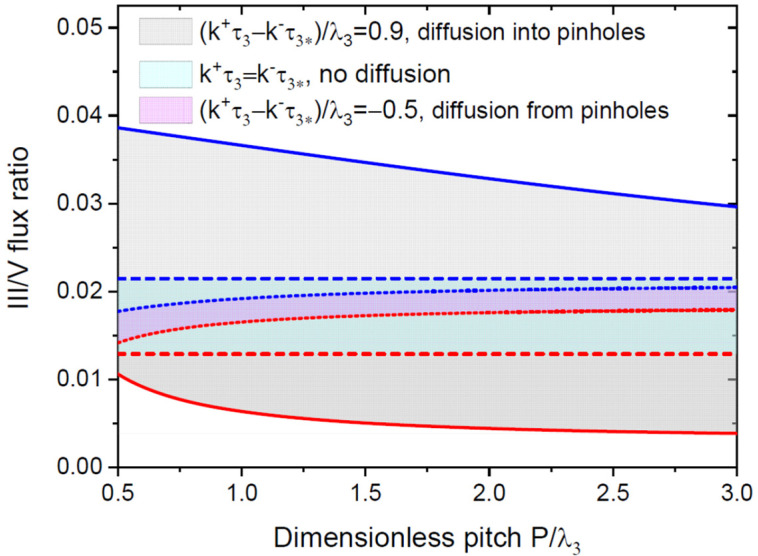
Same as Figure 6 in for the III/V flux ratio versus the dimensionless pitch $P/\lambda_{3\;}$. The dimensionless pinhole radius $R/\lambda_{3\;}$ is fixed at 0.2.

## Data Availability

Not applicable.

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
