# Peer review of "Criterion for Selective Area Growth of III-V Nanowires"

_nanomaterials, 2022, doi:10.3390/nano12203698_

Round 1
Author Response
Please find the response letter in the attachment

Reviewer 2 Report
The manuscript developed a model for selective area growth (SAG) of III-V and III-N nanowires (NWs) in 1D or 2D template geometry. Compared to previous reports, the SAG windows have been plotted in terms of group III and group V fluxes, temperature, and template geometry. Such effort is valuable for the designed growth of nanowires. Nevertheless, some reversions are suggested to make this manuscript easily understandable. As a result, the manuscript in the present form does not look convincing, perhaps requiring at least a careful revision.
The following questions/comments should be addressed essentially:
- Most of the second section “SAG criterion” are quoted from literature, so it doesn’t have to be so verbose.
- The difference between this manuscript and previous reports is that more factors are considered, including the minimum chemical potential required for nucleation, and the influence of group V element on nucleation of III-V crystallites. While, the importance of these two factors has not been fully demonstrated, and it is necessary to be supplemented.
- Most of the figures in the fourth section are fitting curves, and the fitting method needs to be shown in the manuscript to prove its rationality.
- Most of the explanations in the manuscript are derived from mathematical formulas. What are the physical implications of the effects on flow, temperature, and geometric structure? For example, does temperature affect the contact energy between substrate and material? Purely from a mathematical point of view, it seems to be not convincing, because the factors considered in the formula may not be complete. Some descriptions of nucleation dynamics are suggested to be supplemented.
- As mentioned in the manuscript, “all the results can easily be reformulated for SAG by metal-organic chemical vapor deposition (MOCVD) and other vapor deposition techniques”. What is the basis for this claim? Different deposition methods have different principles. For example, MBE can ensure the uniformity of precursor concentration on the substrate surface, while CVD is difficult to achieve.
Author Response

(The authors gave the same response as above.)

Reviewer 3 Report
The manuscript V. G. Dubrovskii is about the modeling of the nucleation of the III-V and III-N NWs. Although the topic is relevant to ongoing III-V research, the results presented here are of minor importance to this field. My concern is that this is a repetition of the papers published by the author, as seen from the self-cited 12 papers. Therefore, I find this manuscript not appropriate for publication in nanomaterials or in any other journal.
Author Response

(The authors gave the same response as above.)

Reviewer 4 Report
In the manuscript entitled "Criterion for selective area growth of III-V nanowires", V.G. Dubrovskii has report a developed model for nucleation of vertical or planar III-V and III-nitride nanowires (NWs) in selective area growth (SAG) on masked substrates with regular arrays of openings.
First of all, the manuscript is not well written, the English style is very poor, and the author often describe using the first singular person.
The manuscript layout is wrong; the author should use paragraphs as introduction, material and methods, results and discussion and conclusion.
In the overall manuscript the author should refer to III-nitride materials and not "III-N".
In the introduction, the author should describe the used SAG approach, reporting the advantages and disadvantages respect to the common ones; moreover, the other cited methods, such as MOCVD, vapour deposition should be briefly described.
The overall manuscript is poor comprehensive; the motivation, objectives and results are not clear.
I cannot accept this manuscript for publication; I reject it.
Author Response

(The authors gave the same response as above.)

Round 2
Reviewer 2 Report
I have no comments.
Reviewer 3 Report
The author has responded to my comment adequately. Although I appreciate the author's insightful comments, the physical concept of the response and the explanations in the manuscript can also be elucidated from the previous papers and even from textbooks. Thus, as an experimentalist in this field, I do not see significant news or a considerable contribution to the field in the present work. I have carefully read previous papers from the author and collaborators, which are interesting combinations of theory and experiments. Unfortunately, the present work falls below the quality of previous works regarding scientific findings. I can not recommend its publication in Nanomaterials. Nevertheless, since I approach mainly from an experimental standpoint, I would recommend inviting a reviewer from theoretical solid-state physics to provide a comprehensive and balanced assessment. Nevertheless, since I approach this mainly from an experimental standpoint, I would recommend inviting a reviewer from theoretical solid-state physics to provide a comprehensive and balanced assessment.
Reviewer 4 Report
The revised version of the manuscript is now acceptable for publication.